# Antialcohol and Hepatoprotective Effects of Tamarind Shell Extract on Ethanol-Induced Damage to HepG2 Cells and Animal Models

**DOI:** 10.3390/foods12051078

**Published:** 2023-03-03

**Authors:** Shao-Cong Han, Rong-Ping Huang, Qiong-Yi Zhang, Chang-Yu Yan, Xi-You Li, Yi-Fang Li, Rong-Rong He, Wei-Xi Li

**Affiliations:** 1College of Traditional Chinese Medicine, Yunnan University of Chinese Medicine, Kunming 650500, China; 2Guangdong Engineering Research Centre of Chinese Medicine & Disease Susceptibility, Jinan University, Guangzhou 510632, China

**Keywords:** alcohol liver disease, chick embryo, tamarind shell extract, oxidative stress, NRF2

## Abstract

Alcohol liver disease (ALD) is one of the leading outcomes of acute and chronic liver injury. Accumulative evidence has confirmed that oxidative stress is involved in the development of ALD. In this study, we used chick embryos to establish ALD model to study the hepatoprotective effects of tamarind shell exttract (TSE). Chick embryos received 25% ethanol (75 μL) and TSE (250, 500, 750 μg/egg/75 μL) from embryonic development day (EDD) 5.5. Both ethanol and TSE were administrated every two days until EDD15. Ethanol-exposed zebrafish and HepG2 cell model were also employed. The results suggested that TSE effectively reversed the pathological changes, liver dysfunction and ethanol-metabolic enzyme disorder in ethanol-treated chick embryo liver, zebrafish and HepG2 cells. TSE suppressed the excessive reactive oxygen species (ROS) in zebrafish and HepG2 cells, as well as rebuilt the irrupted mitochondrial membrane potential. Meanwhile, the declined antioxidative activity of glutathione peroxidase (GPx) and superoxide dismutase (SOD), together with the content of total glutathione (T-GSH) were recovered by TSE. Moreover, TSE upregulated nuclear factor erythroid 2—related factor 2 (NRF2) and heme oxyense-1 (HO-1) expression in protein and mRNA level. All the phenomena suggested that TSE attenuated ALD through activating NRF2 to repress the oxidative stress induced by ethanol.

## 1. Introduction

Alcohol abuse is becoming a global health problem. The immediate consequence of acute drinking is the hangover, with lethargy and cognitive impairment as the most prominent manifestations [1]. Because the liver is responsible for the metabolism of over 90% of consumed ethanol, alcohol liver disease (ALD) has been a main reason of acute and chronic liver injury [2]. For a start, ethanol is converted to acetaldehyde in hepatocytes, which is catalyzed by alcohol dehydrogenase (ADH) and microsomal ethanol-oxidizing system, particularly cytochrome P450 2E1 (CYP2E1) and CYP3A [3,4]. Acetaldehyde is subsequently transformed to acetate, under the catalysis of aldehyde dehydrogenase (ALDH) [5]. Excessive or chronic alcohol consumption causes alcohol metabolism disorder in hepatocytes decreasing ADH and ALDH and increasing CYP2E1. The oxidation of ethanol catalyzed by CYP2E1 accelerates the accumulation of acetaldehyde and produces large amounts of reactive oxygen species (ROS), such as singlet oxygen radicals, superoxide radicals, hydroxyl radicals and hydrogen peroxide [6]. Superfluous ROS combined with disrupted antioxidative defense systems, oxidative stress occurs and results in liver injury, including ethanolic hepatitis, liver fibrosis, cirrhosis, and even liver cancer. However, the treatment options of ALD are still limited. The demand of potential therapies for ALD is urgent.

Tamarind (*Tamarindus indica L.*), also known as Suanjiao, Suandou, Mahanghuang (Dai language), belongs to the subfamily Caesalpinioideae of the family Leguminosae (Fabaceae). It is widely cultured in Africa, Southeast Asia, and South America. Tamarind is high yielding, with an annual output of 300,000 tons in India alone [7]. Tamarind shells are leftovers of direct consumption and by-products of processing. With large amounts, tamarind shells waste resources and pollute the environment. Waste products such as peel and shell contain high levels of polyphenols, flavonoids, anthocyanins, vitamin C, and carotenoids [8], which are expected to be used as a natural source of medicinal antioxidants, cosmetics and food to increase economic benefits. Tamarind shell extract was also reported to attenuate carbon tetrachloride-induced liver injury in mice [9]. Our previous study has demonstrated that tamarind shell extract was rich in flavonoids. In our previous analysis by ultraperformance liquid chromatography-mass spectrometry (UPLC/MS), the major components of TSE were identified as naringenin, luteolin, myricetin, morin, eriocitrin, apigenin, (+) catechin, and taxifolin. We also found that TSE and the main flavonoids had remarkable antioxidant effects in vitro and in vivo [10]. However, limited research was found concerning the pharmacological effects of TSE on liver damage caused by alcohol. Therefore, in this study, the hepatoprotective effects of TSE were studied on ethanol-stimulated HepG2 cells in vitro and ethanol-induced liver injury in zebrafish and chicken embryos. The related mechanism was further explored. The results of this work would hopefully introduce new uses of tamarind shell and promote the comprehensive utilization of tamarind.

## 2. Materials and Methods

### 2.1. Chemicals and Reagents

From Jiangmen Yujun Trading Co., Ltd. (Jiangmen, China), 95% edible ethanol was. Then, 3-(4,5-dimethylthiazol-2-yl)-2,5-diphenyltetrazolium bromide (MTT) was purchased from Sigma (St. Louis, MO, USA). Rhodamine 123 and assay kits for the determination of reactive oxygen species (ROS), total glutathione, superoxide dismutase (SOD), catalase (CAT) and lipid peroxidation (MDA) were procured from Beyotime Biotechnology Company (Shanghai, China). Alcohol dehydrogenase (ADH) and acetaldehyde dehydrogenase (ALDH) assay kits were bought from Solarbio Biotechnology Company (Beijing, China). Triglyceride (TG) assay kits was obtained from Nanjing Jiancheng Bioengineering Research Institute (Nanjing, China). A chicken cytochrome P450 2E1 (CYP2E1) ELISA kit was purchased from Shanghai Jingkang Bioengineering Co., Ltd. (Shanghai, China). Primary antibodies of NRF2 and HO-1 were purchased from Proteintech Group (Chicago, IL, USA). HRP-labeled fluorescent secondary antibody (antimouse and antirabbit) and anti-β-actin were procured from Fude Biological Technology (Hangzhou, China). Dulbecco’s modified Eagle’s medium (DMEM), fetal bovine serum (FBS), and penicillin streptomycin (PS) were purchased from Life Technologies Corporation (Gibco, Grand Island, NY, USA).

### 2.2. Preparation of Tamarind Shell Extract (TSE)

Tamarind fruits (*Tamarindus indica* L.) were purchased in Xishuangbanna (China) and authenticated by Professor Ya-qiong Li, Yunnan University of Chinese Medicine. The fruit shells were separated, dried, and powdered. A total of 500 g of tamarind shell fine powder was refluxed twice with 1500 mL of 95% ethanol for 3 h. The extract was concentrated under reduced pressure, then freeze-dried and stored at 4 °C.

TSE was dissolved in ultrapure water (50 mg/mL) and stored at 4 °C. The solution was diluted to the experimental concentration with the corresponding culture medium in the following experiment.

### 2.3. Zebrafish Maintenance and Exposure

Zebrafish (wild-type AB) were bought from the China zebrafish resource center (Wuhan, China) and maintained following light and temperature standard conditions [11]. The embryos were collected from breeding groups of a 1:1 or 2:1 female-to-male ratio. The fertilized and normally developed eggs were selected and incubated in embryo medium. Zebrafish larvae of five days postfertilization (dpf) were used for subsequent experiments. All procedures were reviewed and approved by the Animal Care and Welfare Committee of Yunnan University of Chinese Medicine (approval No: R-062019009; approval date: 7 March 2019).

#### 2.3.1. Toxicity

The toxicity of TSE to zebrafish larvae was determined by exposing zebrafish to TSE. Briefly, 5 dpf zebrafish larvae were divided to TSE exposure groups at different concentrations and, control group at random. TSE was added to the larvae (5, 10, 20, 40, and 60 μg/mL). The control group received embryo culture solution. Exposures were performed in triplicate. The survival rate was recorded every day until 9 dpf.

#### 2.3.2. Exposure

The exposure of zebrafish to ethanol was following the protocol described earlier [12] with some modifications. In brief, at 5 dpf, zebrafish larvae were randomised into four groups, which were control group, model group, and two TSE treatment (10 μg/mL and 20 μg/mL) groups, and cultured in a six-well plate (50 fish per well, three multiple wells per group). All larvae except the control group were exposed to 1.5% ethanol or 1.5% ethanol with TSE prepared in embryo culture medium. Fish in control group were given embryo culture solution. The zebrafish were incubated for 32 h at 28.5 °C. Six zebrafish were selected from each group for phenotype analysis. The remaining fish were transferred to a 12-well plate (30 fish per well) and added 3 mL of appropriate solution: the experimental groups received TSE (10 μg/mL and 20 μg/mL, diluted in embryo culture solution), whereas control and model groups received embryo culture solution. After incubation for 48 h, the fish were collected for subsequent experiments. Triplicate wells were set for each group.

### 2.4. Chicken Embryos and Treatment

The fertilized Leghorn eggs required for the experiment were obtained from the poultry farm of South China Agriculture University (Guangzhou, China). The eggs were incubated in an incubator (ProCon automatic systems GmbH & Co. KG, Luebeck, Germany) at 38 °C under 70% humidity. At the embryonic development day (EDD) 5.5, live fertilized eggs were selected by an egg illuminator marked the air chamber, and a small hole was introduced into the shell for drug administration. Live eggs were randomly divided into a control group, an ethanol group and TSE (250, 500 and 750 μg per egg) groups with 24 eggs in each group. The eggs in control group were treated with 75 μL bird saline per egg, the eggs in ethanol group were administrated with 75 μL 25% ethanol per egg, and the eggs of TSE groups were dosed with 75 μL indicated dosage of TSE coadministration with 25% ethanol. The eggs were administered with appropriate solution every two days from EDD 5.5 until the EDD 15. The embryos were collected for subsequent experiments. A schematic illustration of the above protocol is presented in Figure 1.

### 2.5. Cell Culture

Human hepatoma HepG2 cells (Procell CL-0103) were bought from Punuosai Life Technology (Wuhan, China). HepG2 cells were cultured in DMEM supplemented with 12% FBS and 1% PS in an incubator (Thermo, Waltham, MA, USA) at 37 °C, 5% CO_2_ and 65% humidity. The following experiments were completed by using the third- to tenth-generation cells.

### 2.6. Cell Toxicity and Cell Experiment

The cytotoxicity of TSE and ethanol on HepG2 cells were determined by MTT assay. In brief, HepG2 cells were seeded into 96-well plates (2 × 10^4^/well) and cultured overnight. Afterward, the cells were treated with TSE or ethanol for 24 h, and the cell toxicity was determined by MTT assay.

To assess the protective effect of TSE against ethanol-induced injury, HepG2 cells were grown in 48-well plates at a density of 6 × 10^4^/well for 12 h. After treatment of TSE (5, 10, 20, and 40 μg/mL) for 24 h, cells were administrated with ethanol (700 mM) for 3 h. Cell viability was assessed by MTT assay. The DCFH-DA assay was carried out to quantified the generation of intracellular ROS on a flow cytometry (Beckman Coulter, Bria, CA, USA) as reported by Zhang et al. [13]. The mitochondrial membrane potential of HepG2 cells was measured by using rhodamine 123 as a fluorescent probe (Beyotime, Shanghai, China), following the method reported by Wang et al. [14].

### 2.7. Zebrafish Behavioral Test and Evaluation of ROS Level in Zebrafish

Zebrafish were treated as mentioned above. Five zebrafish were randomly selected from each group and placed in a 96-well plate individually. The swimming behavior, including total moving distance, trace, and total activity time, was monitored every 5 s for a total of 70 min by using a zebrafish behavior automatic analyzer (View Point Behavior Technology, Lyon, France) according to the manufacturer’s guide.

For the measurement of ROS, zebrafish were homogenized (Beyotime, Shanghai, China) in PBS on ice, then centrifuged at 12,000 rpm for 15 min at 4 °C. A BCA protein assay kit was utilized to quantify the protein concentration. The protein was added to a black 96-well plate (55.84 μg in 50 μL/well) containing DCFH-DA fluorescent probe (50 μL/well, final concentration 10 μM). The intensity of fluorescence was recorded in a Synergy H1 fluorescence microplate reader (BioTek, Winooski, VT, USA), the excited and emitted wavelength were 488 and 525 nm.

### 2.8. Histological Analysis

Liver damage was evaluated by histological examination in the chick embryos, as the method described by Zhang et al. and with some modifications [13]. In brief, chicken embryo liver tissue was fixed in 4% paraformaldehyde solution for 10 days and then embedded in paraffin, then sectioned into 5 μm paraffin slices. After dewaxing and rehydrating, the slice was stained with hematoxylin and eosin (H&E) and sirius red (Servicebio, Wuhan, China). The morphology of the tissue sections was observed under an automatic scanning microscope (Pannormic, Budapest, Hungary).

### 2.9. Biochemical Analysis

The chicken embryo liver tissue homogenate was prepared on ice and then centrifuged at 12,000 rpm for 10–20 min at 4 °C. The protein content of supernatant was detected with BCA protein assay kit, and the supernatant was collected for subsequent experiments. The hepatic level of MDA, T-GSH and TG and the activities of GPx, SOD, CAT, ADH, ALDH and CYP2E1 were assessed by commercial kits according to the manufacturers’ protocols.

### 2.10. Western Blot Analysis

Total proteins of HepG2 cells and chicken embryo liver tissues were extracted, respectively, as described before. The concentration of protein was determined with BCA protein assay kit. Protein samples (25 μg) were loaded on 10% SDS-PAGE to separate and blotted to a PVDF membrane. The transferred PVDF membrane was sealed in 5% skimmed milk solution at 25 °C for 2 h after being cleaned with TBST, then incubated with primary antibodies to detect NRF2 (1:1500) and HO-1 (1:1500) at 4 °C overnight; β-actin (1:5000) served as an internal standard. Subsequently, the membranes were incubated with HRP-labeled fluorescent secondary antibody, antimouse or antirabbit (1:5000) at room temperature for 2 h. Finally, the protein bands were imaged by using an ECL Detection Kit (Fdbio science, Hangzhou, China) and the Vilber FUSION FX 6 imaging system (Vilber, Collégien, France).

### 2.11. qRT-PCR

Total RNA was isolated from chick embryo liver tissues, HepG2 cells, and zebrafish by using TRIzol Reagent as described by the manufacturer’s instructions. A NanoDrop ND-2000C spectrophotometer (Thermo, Waltham, MA, USA) was used to quantify the concentration of RNA. A total of 1000 ng of the total RNA was utilized to synthesize cDNA following the manufacturer’s guide (TransGen Biotech, Beijing, China). The result of qRT-PCR was analyzed by LightCycler 96 system (Roche, Switzerland) by using the TransStart Top Green qPCR Supermix Kit (TransGen Biotech, Beijing, China). The target gene expression was assessed by using the 2 ^−ΔΔCT^ method and GAPDH was used as the housekeeper gene and to a control group sample. The sequences of primers were listed in Table 1.

### 2.12. Statistical Analysis

All values were expressed as mean ± SD. Data were analyzed by one-way variance (ANOVA) followed by Tukey’s multiple comparison with SPSS 25 software (IBM SPSS Statistics 25.0, Armonk, NY, USA). The *p* < 0.05, *p* < 0.01 and *p* < 0.001 were set at the threshold for statistical significance, high statistical significance, and very high statistical significance, respectively. Graphs were constructed using GraphPad Prism 8 (GraphPad Prism 8.0, San Diego, CA, USA) and Adobe illustrator CS6 software (Adobe illustrator CS6, San Jose, CA, USA).

## 3. Results

### 3.1. TSE Protected Ethanol-Induced Damage in HepG2 Cells

The cell toxicity of TSE was tested on HepG2 cells firstly. The cell viability suggested that TSE had no significant toxicity to HepG2 cells with a concentration of less than or equal to 60 μg/mL (Figure 2A). When treated with ethanol, the cell viability dropped significantly (Figure 2B,C), accompanied by evident morphology changes, reflected as gradual shrinkage, increased intercellular gaps and complete loss of the original morphological characteristics (Figure 2D). We found that all doses of TSE could rescue the ethanol-treated cells from death and reverse the morphological alterations (Figure 2C,D). Numerous studies have confirmed that the oxidation of ethanol produces large amounts of ROS and results in the impairment of mitochondrial [17]. Using fluorescence probes, we measured the intracellular ROS level and the mitochondrial membrane potential. Similarly, TSE effectively inhibited the accumulation of ROS (Figure 2E), and restored the disrupted mitochondrial membrane potential in HepG2 cells induced by ethanol (Figure 2F). These findings indicated that TSE was effective on preventing ethanol-induced cell damage, eliminating intracellular ROS and reversing the depolarization of mitochondrial membrane.

### 3.2. TSE Ameliorated Ethanol-Induced Behavior Changes and Oxidative Stress in Zebrafish

To further study the protective effect against ethanol of TSE in vivo, the ethanol-exposed zebrafish model was employed. At first, we evaluated the toxicity of TSE in zebrafish. The survival rate in all groups exceeded 91.67% (96 h after administration), indicating that TSE had negligible toxicity with the concentration of 60 μg/mL (Figure 3A). As shown in Figure 3B, ethanol produced severe developmental abnormality of zebrafish, presented as spinal curvature, pericardial edema, hepatomegaly yolk sac edema and lack of swim bladder. TSE prevented the incidence of malformation induced by ethanol. In the motor activity test, ethanol caused a dramatic reduction in total swimming distance and the amount of active time compared to the untreated fish (*p* < 0.01) (Figure 3C). TSE (20 μg/mL) greatly increased the swimming distance and the amount of active time. Furthermore, we detected the content of ROS in zebrafish. Consistently, TSE strikingly blocked the excessive ROS generated by ethanol (Figure 3D). These data illustrated that TSE relieved alcoholism and suppressed ROS produced by ethanol.

### 3.3. TSE Ameliorated Ethanol-Provoked Liver Dysfunction and Alleviated Liver Injury in Chicken Embryos

As the major site of detoxification, liver is susceptible to stimuli and pollutants. Furthermore, the lipase liberated by ethanol metabolism catalyzed the production of TG and the deposition of fat in liver cells [18]. Thus, TG is one of the biomarkers of ethanol-induced liver damage [19]. After ethanol administration, the chicken embryo survival rate dropped to 70.83%, whereas TSE treatment improved the survival rate (Figure 4A). In addition, TSE reversed the increased liver index caused by ethanol (Figure 4B). We also noted a significant increment of TG levels in embryo liver. The elevated TG levels could be remarkably decreased by TSE (Figure 4C). Ethanol consumption also gives rise to the accumulation of ROS disrupted metabolic enzymes. All of these changes trigger pathological responses, which contribute to the etiology of ethanol-induced liver injury [20,21]. Ethanol impeded the development of embryo liver, which was similar to the influence on zebrafish larvae. In contrast with the normal embryo liver, the ethanol group showed obvious hemorrhage and fatty liver (Figure 4D). As presented in Figure 3F, hepatocytes in the control group were complete, with a neat, tight arrangement, intact nucleus, and clear gaps between hepatic sinuses. In contrast, ethanol led to evident hepatic pathological alterations. The liver sinuses dilated with disordered cell arrangement. Hemorrhage, inflammatory infiltration, fat deposition, and fibrosis were also found. TSE mitigated fat vacuole formation, inflammatory infiltration, and fibrosis. (Figure 4E,F).

### 3.4. TSE Corrected the Disorder of Ethanol-Metabolic Enzymes In Vivo

Disturbing metabolic enzymes are a typical feature of ethanol-induced liver damage. Ethanol is firstly transformed to acetaldehyde mainly by ADH, CYP2E1, and CYP3A. Acetaldehyde is subsequently metabolized by ALDH. Needless ROS generated during ethanol oxidation, especially through CYP2E1 [5]. Additionally, as an inducible enzyme, the expression of CYP2E1 is enhanced by alcohol ingestion, which exacerbates the production of ROS. Meanwhile, ethanol counteracts the effect of ALDH, leading to the accumulation of acetaldehyde, a robust hepatotoxic [12,18,22].

To evaluate the recovering effect of TSE on ethanol-evoked liver damage, we detected the expressions of ethanol metabolism-related genes by qPCR. In Figure 5A, we observed a slight rise in the mRNA expression of ethanol metabolism-related enzyme *cyp2y3* in ethanol-treated zebrafish. The mRNA expressions, *cyp3a65*, *adh8a*, and *adh8b* in zebrafish were markedly upregulated following ethanol exposure (Figure 5A). All doses of TSE could greatly abrogate the expressions of *cyp3a65*, *adh8a,* and *adh8b* in ethanol-stimulated zebrafish. In ethanol-exposed chick embryo liver, as shown in Figure 5B, the mRNA expression of *Cyp3a4*, *Cyp3a7*, and *Cyp2d6* were upregulated in the ethanol group, and these genes were involved in ethanol oxidation [23]. Similarly, the expression of *Cyp2c45* mRNA related to fat synthesis [24] was also upregulated by alcohol. TSE treatment decreased the expression of these genes. In addition, CYP2E1 enzyme activity was significantly increased. The activity of ALDH was drastically inhibited, accompanied by a mildly ADH. Likewise, TSE inhibited the overactive CYP2E1 and restored the inactive ALDH (Figure 5C–E). TSE was capable of correcting the disorder of ethanol-metabolic enzymes.

### 3.5. TSE Attenuated Oxidative Stress in Ethanol-Induced Chick Embryo Livers

Ethanol exposure can interfere with the antioxidant system that protects hepatocytes against ROS damage. Antioxidant defense systems within cells include SOD, GPx, CAT, and T-GSH, which are important in defending cells from oxidative damage and preventing lipid peroxidation. MDA is the final metabolite of lipid peroxidation and play an indicative function in oxidative damage. [25,26]. We noted that ethanol disrupted the antioxidant defense system of the liver, with declined SOD, GPx, and CAT activities as well as the T-GSH content (Figure 6A–D), although the change of CAT activity was not significant. In parallel, the MDA level in embryo liver was elevated after ethanol treatment (Figure 6E). Administration of TSE at 250, 500 or 750 μg/egg evidently recovered the activities of SOD. Except for 250 μg/egg of TSE, the content of T-GSH was also evidently increased. Only 750 μg/egg of TSE can significantly restore the activity of GPx and CAT. Of course, the production of MDA was also inhibited by TSE.

### 3.6. TSE Activated the Nuclear Factor Erythrocyte-2-Related Factor 2 (NRF2)-Mediated Antioxidant Response

NRF2 is a vital player in regulating intracellular redox homeostasis in cells [27]. Normally, NRF2 exists in the cytoplasm. When oxidative stress occurs, NRF2 translocates to the nucleus and binds antioxidant response elements (ARE) to drive the expression of the downstream genes, such as NQO-1, HO-1, SOD, GPx, CAT, and GSH [20,28]. Improved antioxidant capacity can restrain ROS-induced liver damage. However, chronic alcohol ingestion inhibits the function of NRF2, exacerbating oxidative stress in the liver [18,29]. Therefore, we monitored the levels of NRF2 and HO-1 both in chicken embryo liver and HepG2 cells. In chicken embryo liver tissue, NRF2 and HO-1 were significantly downregulated at protein and mRNA levels in ethanol group, whereas remarkable restoration was detected in the TSE treatment groups (Figure 7A,B). The protein levels of NRF2 and HO-1 were evidently suppressed in ethanol-stimulated HepG2 cells, accompanied by mRNA expressions of *NRF2* and *HO-1* significant alteration. Pretreated with TSE (5 μg/mL, 10 μg/mL and 20 μg/mL) significantly restored the protein levels of ethanol-inhibited NRF2 and HO-1 (Figure 7C). The mRNA levels of *NRF2* and *HO-1* were also apparently raised in HepG2 cells by pretreated with 5 μg/mL and 10 μg/mL TSE (Figure 7D). Those findings elucidated the fact that TSE alleviated ethanol-induced liver damage by modulating NRF2 and its downstream antioxidant enzymes.

## 4. Discussion

Until now, common treatments for ALD have been nutritional support and liver protection therapy, but we still lack efficacious and targeted treatment options. Suitable models for ALD are urgently needed to explore the pathogenesis and therapeutic strategies. Current experimental models of ALD are mainly based on rodents and cell lines. However, ALD related experiments on in vitro systems have limits in mimicking the symptoms of entire living organisms. In vivo ALD studies performed on rodents are of high cost, have long cycles, and have complicated operations with ethical problems. In contrast, chicken embryos are easy to operate, and the developmental process is easy to observe. In this work, we built an ethanol-induced liver injury model on chicken embryos for the first time. The embryonic liver of chicken has been formed at 20–22 somite stage (50–53 h after hatching) [30,31], and hemoblast cells have appeared in the hepatic sinus space at EDD 5 [32]. Therefore, we exposed chick embryos to 75 μL 25% (*v*/*v*) ethanol every two days from EDD5.5 until EDD15. There was a significant increase in the liver index due to ethanol. Severe hypertriglyceridemia is a prevalent complication of alcohol consumption, so TG is an important indicator of ALD [18]. A significant rise of TG in liver was noted after ethanol administration as well. The most important clinical pathological manifestations of ALD were steatosis, hepatocyte damage, and inflammatory infiltration. In ethanol-exposed embryo liver, we found obvious hepatic pathological alterations, presented as hemorrhage, inflammatory infiltration, fatty liver tissue and fibrosis. The disorder of metabolic enzymes was also noticed. Ethanol substantially inhibited the liver ALDH activity leading to excessive accumulation of acetaldehyde. ADH, another metabolic enzyme related to alcohol was mildly suppressed. Meanwhile, the activity of CYP2E1 was enhanced, and CYP3A7, the main CYP3A enzyme in fetal liver [33], was significantly upregulated by ethanol. These CYPs are associated with the generation of large amounts of ROS. Because oxidation stress is linked to alcohol-induced liver disease, we detected the change of hepatic antioxidant defense system in chick embryos subjected to ethanol. The activities of antioxidant enzymes including SOD, GPx, and CAT were decreased together with the content of low molecular weight antioxidant GSH. On the other hand, MDA, a final product and indicator of lipid peroxidation, remarkably elevated. The disrupted antioxidant defense system along with enhanced lipid peroxidation suggested the occurrence of oxidative stress. All the above indicated that the ALD model on chick embryos was successfully established.

Tamarind shells, wastes of consumption and processing, lack proper utilization for a long time. In our previous research, we found that tamarind shell was rich in functional chemicals. By using LC/MS, we identified and quantified eight flavonoids in TSE. These included naringenin, luteolin, myricetin, morin, eriodictyol, apigenin, catechin, and taxifolin. The antioxidative effects of TSE and the eight flavonoids were evaluated in vitro and in vivo [10]. Our research indicated that TSE possessed potent antioxidant capacity both in vitro and in vivo. The tamarind shell extract (60 μg/mL) was low in cytotoxicity, with no influence on the growth of HepG2 cells and zebrafish.

In the current work, we systematically evaluated the hepatoprotective effect of TSE on ethanol-induced liver damage in chick embryos, zebrafish, and HepG2 cells. TSE was found to rescued HepG2 cell from ethanol. In the presence of ethanol, there was a dramatic decrease in the motor activity of zebrafishes. Severe morphology changes were observed in ethanol-exposed zebrafish and HepG2 cells. Combined with the alteration in embryo liver described earlier, ethanol caused severe hepatic damage in vitro and in vivo. TSE could improve the pathological changes induced by alcohol in chick embryo liver, zebrafish, and cells, and increase in the amount of time and total distances of swimming in zebrafish. All the abovementioned results suggested that TSE relieved ethanol-provoked hepatic damage in vitro and in vivo.

Oxidation stress is an important feature of ethanol-induced liver damage. By using the fluorescence probe DCFH-DA, we found excessive ROS in zebrafish and HepG2 cells cultured with ethanol. Superfluous ROS contributes to mitochondrial impairment, including a reduction of the mitochondrial membrane potential (ΔΨm) and damaged the permeability of membrane. The dysfunctional mitochondrial not only produces more ROS, but also releases cytochrome c, and then activates caspase 9 cascade which causes apoptosis of hepatocyte [34]. We noticed an evident reduction of ΔΨm and cell viability in HepG2 cells within ethanol incubation. Furthermore, the disruption of antioxidant defense system was found in ethanol-administrated embryo liver, as manifested by decreased SOD, GPx, CAT, and total GSH and increased MDA. In accordance with our previous research, TSE could efficiently scavenge the overproduced ROS in alcohol-exposed zebrafish and HepG2 cells. The ΔΨm of HepG2 cells was rebuilt by TSE. A salient remission of the malfunctional antioxidant defense system was also noted in chick embryo liver with the treatment of TSE. Taken together, TSE efficaciously combated the oxidation stress triggered by ethanol.

Disturbing metabolic enzymes re another outcome of alcohol intake. ADH, CYP2E1, and CYP3A are responsible for converting alcohol into acetaldehyde. However, CYP2E1-mediated oxidation of ethanol liberates large amounts ROS. Meanwhile, ethanol is a strong inducer of CYP2E1. The elevated CYP2E1 generates more ROS [35]. Additionally, ALDH, a major player in acetaldehyde metabolism, is severely inhibited by ethanol and results in the accumulated acetaldehyde, a powerful hepatotoxin. In chick embryo liver, the activities of ALDH and ADH were hampered by ethanol with the comparison of normal liver. TSE was capable of reversing the metabolic enzyme disorder by recovering ALDH and ADH and blocking CYP2E1 (Figure 8). 

NRF2/ARE pathway is essential in maintaining the redox homeostasis in humans. NRF2 mediates the transcription of antioxidation and detoxification enzyme genes, such as SOD, GPx, CAT, GSH, NQO-1, and HO-1 [36,37]. Thereafter, the oxidative homeostasis is reestablished. It has been reported that flavonoids such as luteolin, apigenin, naringenin and epicatechin are activators of NRF2/ARE in multiple cell lines. Therefore, we proposed that the protection of TSE on ethanol-evoked liver damage was related to the activation of NRF2. Western blot and PCR results indicated the apparent downregulation of NRF2 and HO-1 in ethanol exposed embryo liver. The levels of both genes were also decreased in HepG2 cells. These data combined with the inhibition of antioxidant defense system demonstrated that the NRF2 pathway was suppressed by ethanol. Consistently, TSE could significantly activate NRF2, following upregulation of HO-1 at the mRNA and protein level in embryo liver (Figure 8). Afterward, the oxidative homeostasis was rehabilitated.

## 5. Conclusions

In this study, we successfully built an ALD model on chick embryo to explore the hepatoprotective activity of TSE. TSE effectively attenuated ethanol-induced liver injury by relieving oxidative stress via activating the NRF2 pathway. These findings could facilitate the utilization of TSE, increase economic benefits, and provide raw material support for the study of natural ingredients of antialcohol and liver protection drugs. In addition, chicken embryo has the characteristics of simple and easy access, short experimental period and low experimental cost. This model of chicken embryo alcoholic liver injury can be used for high-throughput evaluation and screening of more active substances for antialcohol and liver protection.

## Figures and Tables

**Figure 1 foods-12-01078-f001:**
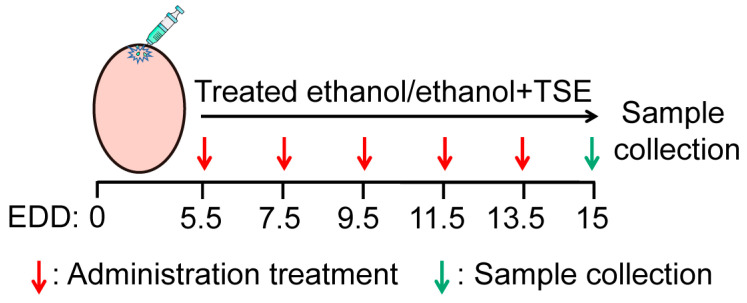
Schematic diagram of chicken embryo treatment protocol.

**Figure 2 foods-12-01078-f002:**
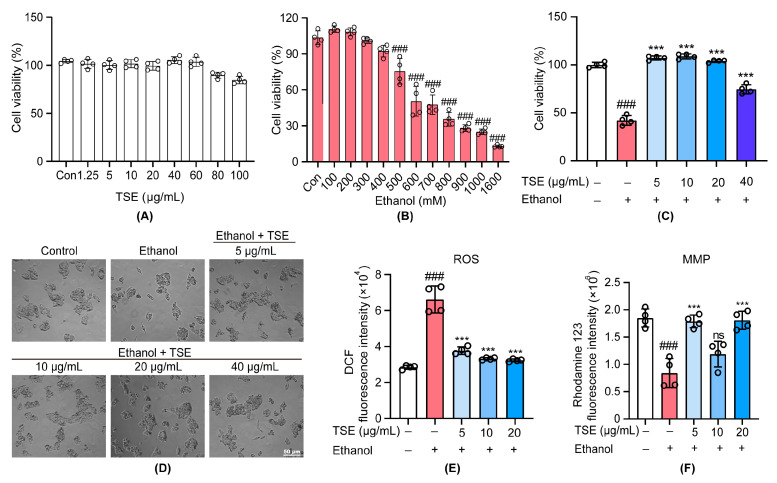
TSE protected HepG2 cells against ethanol-induced damage. (**A**) MTT assay of HepG2 cell viability following treatment with TSE for 24 h, *n* = 4 biologically independent experiments. (**B**) MTT assay of HepG2 cell viability following treatment with ethanol for 24 h, *n* = 4 biologically independent experiments. (**C**) MTT assay of HepG2 cell viability following pretreatment with or without TSE for 24 h before cotreatment with or without ethanol (700 mM) for 3 h, *n* = 4 biologically independent experiments. (**D**) Morphology of cells under microscope (magnification ×200). (**E**) Flow cytometric analysis of intracellular ROS levels detected by DCFH-DA probe, *n* = 4 biologically independent experiments. (**F**) Flow cytometric analysis of mitochondrial membrane potential by rhodamine 123 staining, *n* = 4 biologically independent experiments. Data represent the mean ± standard deviation and significant differences were analyzed by one-way ANOVA. ^###^
*p* < 0.001 vs. control group; *** *p* < 0.001 vs. ethanol group. ns, not significant vs. ethanol group.

**Figure 3 foods-12-01078-f003:**
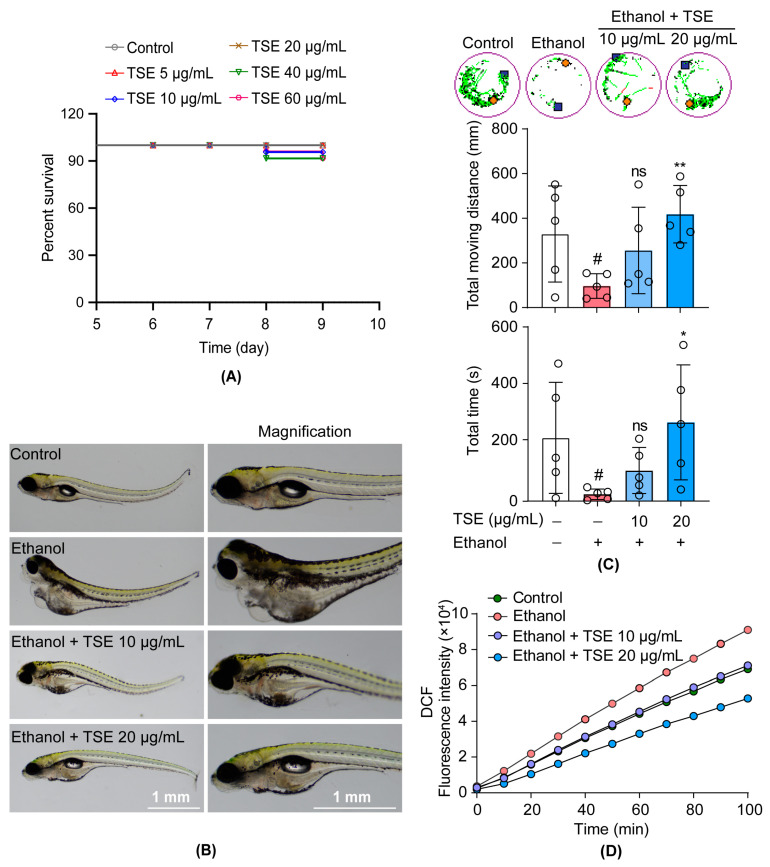
TSE ameliorated ethanol-induced behavior changes and oxidative stress in zebrafish. (**A**) TSE was used to treat 5 dpf zebrafish for 96 h to detect the toxicity of TSE. (**B**) TSE resisted ethanol-induced phenotypic changes in zebrafish, scale bar, 1 mm. (**C**) TSE ameliorated ethanol-induced behavior changes and increased the movement distance and activity time, *n* = 5 biologically independent samples. (**D**) The effect of TSE on the time-dependent changes of ethanol-induced DCF fluorescence intensity. Data represent the mean ± standard deviation and significant differences were analyzed by one-way ANOVA. ^#^
*p* < 0.05 vs. control group; * *p* < 0.05, ** *p* < 0.01 vs. ethanol group; ns, not significant vs. ethanol group.

**Figure 4 foods-12-01078-f004:**
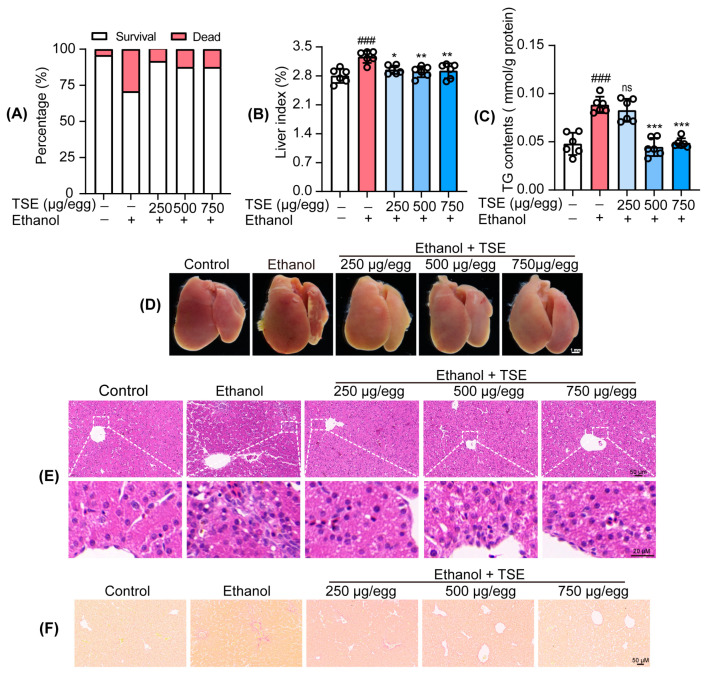
TSE ameliorated ethanol-induced chicken embryo death and liver damage. (**A**) TSE reversed ethanol-induced chick embryo death rates and (**B**) liver index, *n* = 6 biologically independent samples. (**C**) The effect of TSE on ethanol-induced changes of TG level in chicken embryo liver, *n* = 6 biologically independent samples. (**D**) Representative images of chicken embryo liver; scale bar, 1 mm. (**E**) H&E staining showing the histopathological changes of livers; scale bar, 50 μm (up), 20 μm (bottom). (**F**) Sirius red staining showing liver fibrosis; scale bar, 50 μm. Data represent the mean ± standard deviation and significant differences were analyzed by one-way ANOVA. ^###^
*p* < 0.001 vs. control group; * *p* < 0.05, ** *p* < 0.01, *** *p* < 0.001 vs. ethanol group; ns, not significant vs. ethanol group.

**Figure 5 foods-12-01078-f005:**
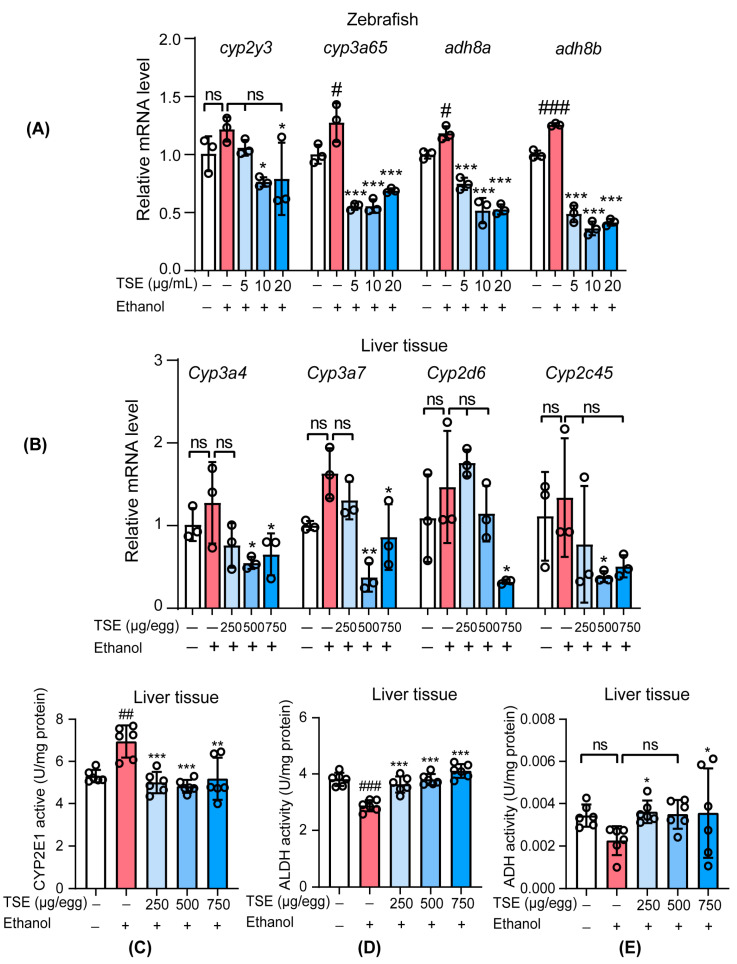
TSE ameliorated ethanol-induced metabolism dysregulation in vivo. (**A**) qRT-PCR analysis of the mRNA levels of *cyp2y3*, *cyp3a65*, *adh8a,* and *adh8b* in zebrafish, *n* = 3 biologically independent samples. (**B**) qRT-PCR analysis of the mRNA levels of *Cyp3a4*, *Cyp3a7*, *Cyp2d6,* and *Cyp2c45* in liver tissue, *n* = 3 biologically independent samples. Activity of (**C**) CYP2E1, (**D**) ALDH, and (**E**) ADH in the chicken embryo liver tissues, *n* = 6 biologically independent samples. Data represent the mean ± standard deviation and significant differences were analyzed by one-way ANOVA. ^#^
*p* < 0.05, ^##^
*p* < 0.01, ^###^
*p* < 0.001 vs. control group; * *p* < 0.05, ** *p* < 0.01, *** *p* < 0.001 vs. ethanol group; ns, not significant vs. ethanol group.

**Figure 6 foods-12-01078-f006:**
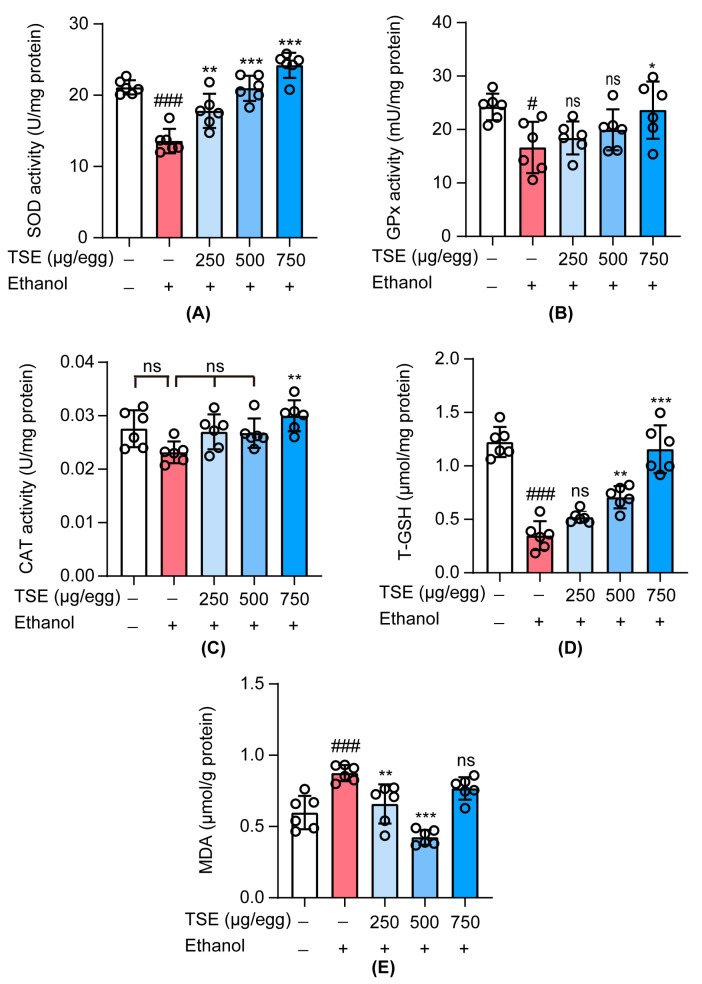
The antioxidant effect of TSE on ethanol-induced chicken embryo liver damage. (**A**–**E**) are the effects of TSE on the changes of ethanol-induced (**A**) SOD, (**B**) GPx, (**C**) CAT, (**D**) T-GSH, and (**E**) MDA levels in ethanol-induced chicken embryo liver, *n* = 6 biologically independent samples. Data represent the mean ± standard deviation and significant differences were analyzed by one-way ANOVA. ^#^
*p* < 0.05, ^###^
*p* < 0.001 vs. control group; * *p* < 0.05, ** *p* < 0.01, *** *p* < 0.001 vs. ethanol group; ns, not significant vs. ethanol group.

**Figure 7 foods-12-01078-f007:**
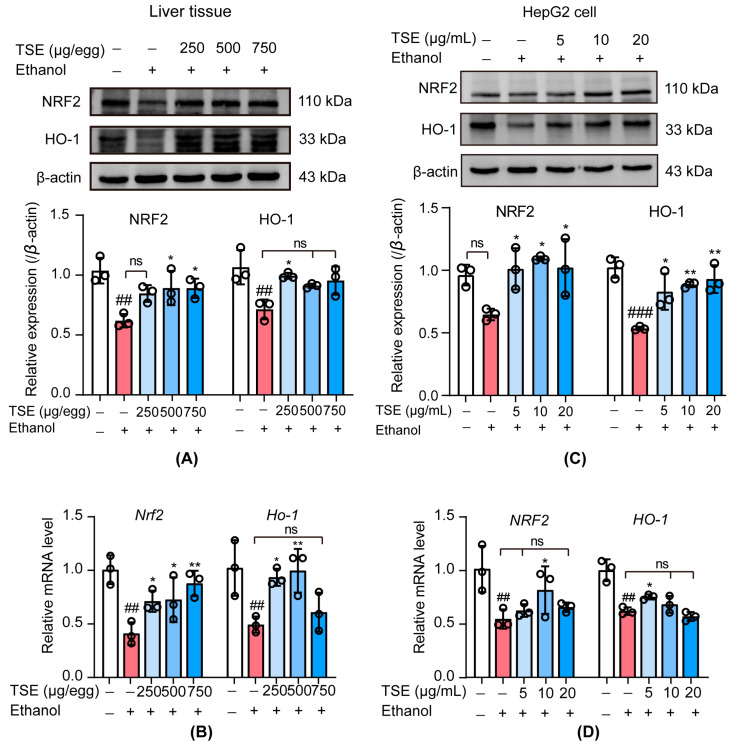
Effect of TSE on NRF2-mediated antioxidant signaling in ethanol-induced liver damage. (**A**) NRF2 and HO-1 protein expression levels in chicken embryo liver tissue, *n* = 3 biologically independent samples. (**B**) qRT-PCR analysis of the mRNA levels of *Nrf2* and *Ho-1* in chicken embryo liver tissue, *n* = 3 biologically independent samples. (**C**) NRF2 and HO-1 protein expression levels in HepG2 cells, *n* = 3 biologically independent experiments. (**D**) qRT-PCR analysis of the mRNA levels of *NRF2* and *HO-1* in HepG2 cells, *n* = 3 biologically independent experiments. Data represent the mean ± standard deviation and significant differences were analyzed by one-way ANOVA. ^##^
*p* < 0.01, ^###^
*p* < 0.001 vs. control group; * *p* < 0.05, ** *p* < 0.01 vs. ethanol group; ns, not significant vs. ethanol group.

**Figure 8 foods-12-01078-f008:**
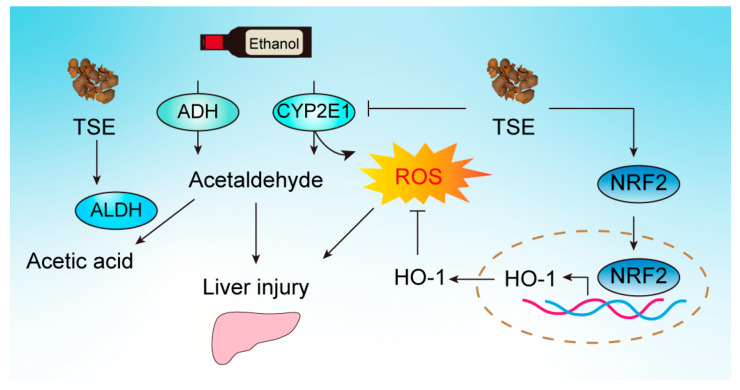
A schematic illustration of tamarind shell extract against ethanol-induced liver damage. Tamarind shell extract can improve alcohol-induced liver injury by improving alcohol metabolism and activating NRF2 pathway to inhibit oxidative stress.

**Table 1 foods-12-01078-t001:** Sequences of primers used for qRT-PCR [12,13,15,16].

Species	Gene	Forward Primer (5′–3′)	Reverse Primer (5′–3′)
Zebrafish	*cyp2y3*	TATTCCCATGCTGCACTCTG	AGGAGCGTTTACCTGCAGAA
Zebrafish	*cyp3a65*	AAACCCTGATGAGCATGGAC	CAAGTCTTTGGGGATGAGGA
Zebrafish	*adh8a*	CGAGTACACCGTCATCAAC	AGCACCGAGTCCGAATAC
Zebrafish	*adh8b*	ATTGATGATGATGCTCCTCTG	TAGACCAACCGCACCAAG
Zebrafish	*gapdh*	TGGTGCTGGTATTGCT	TTGCTGTAACCGAACTCA
Chicken	*Cyp3a4*	TCATAGTGTTGTTCCCCTT	GGTATCCTTCTTCCCGTTC
Chicken	*Cyp3a7*	GACTCCATGAACAACCCCAA	AAATCTACTCTGCCCGTGTG
Chicken	*Cyp2d6*	GAACCCTGCTTACATCCGAGA	CATGAACAGGAACGCCCAT
Chicken	*Cyp2c45*	CGGAGACAACAAGCACCACCA	TTCGTGATCGTCCTACTACCC
Chicken	*Nrf2*	CATAGAGCAAGTTTGGGAAGAG	GTTTCAGGGCTCGTGATTGT
Chicken	*Ho-1*	AACGCCACCAAGTTCAGTCTCC	AGCTTCTGCAGCGCCTCAA
Chicken	*Gapdh*	AGAACATCATCCCAGCGT	AGCCTTCACTACCCTCTTG
Human	*NRF2*	CCTCAACTATAGCGATGCTGAATCT	AGGAGTTGGGCATGAGTGAGTAG
Human	*HO-1*	CCAGTGCCACCAAGTTCAAG	CAGCTCCTGCAACTCCTCAA
Human	*GAPDH*	GCCTCAAGATCATCAGCAATGC	CCTTCCACGATACCAAAGTTGTCAT

## Data Availability

The data presented in this study are available on request from the corresponding author.

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
