# Peer review of "Antialcohol and Hepatoprotective Effects of Tamarind Shell Extract on Ethanol-Induced Damage to HepG2 Cells and Animal Models"

_foods, 2023, doi:10.3390/foods12051078_

Round 1

Reviewer 1 Report

The manuscript entitled: Anti-alcohol and hepatoprotective effects of tamarind shell extract on ethanol-induced damage to HepG2 cells and animal models” is scientifically sound and interesting. The study is well-designed. Please find my comments below that might help to improve the quality of this manuscript.

-   The scientific name (Tamarindus indica) should be in an italic form.

-   2.2. Preparation of tamarind shell extract

500 g of tamarind shell fine powder was 86 refluxed twice with 1500 mL of 95% ethanol for 3 hours. The solvent was removed under 87 reduced pressure. The extract was freeze-dried and stored at 4 °C.

### Just wondering that why did you have to use freeze dryer after evaporation the ethanol by using the rotary evaporator as your extract contained with water only 5% (95% ethanol)!!!?

-   The authors should add the solvent as you used for dissolve your extract in each experiment because it is important for bioassay, sometimes the solvent can give the false positive or false negative result.

-   2.3. Zebrafish maintenance and exposure

### I would suggest to make two subtitles as:

2.3.1 Toxicity

2.3.2 Exposure

In order to clarify the different concentration as you used in these experiments.

-   3.1. TSE protected ethanol-induced damage in HepG2 cells

The cell toxicity of TSE was tested on HepG2 cells firstly. The cell viability suggested that TSE had little toxicity to HepG2 cells with a concentration of £ 60 µg/mL (Figure 1A).

### I think it should be…….. ³ 60 µg/mL (Figure 1A).

-   3.1. TSE protected ethanol-induced damage in HepG2 cells

We found that all doses of TSE could rescue the ethanol-treated cells from death and reverse the morphological alterations (Figure 1C and D)

### How about the dose at 40 µg/mL?

-   Please edit small size of the letter at Figure 5….. n = 6 biologically independent….

-   Page 13, line 409, please edit 60 µg mL-1 to µg/mL

-   References: should follow the journal format.

Cheers

Author Response

We are grateful for your positive comments and constructive feedback. The manuscript has been exactly revised according to your suggestions point by point. All the modifications have been highlighted in red in the revised manuscript.

Comments and response:

  1. The scientific name (Tamarindus indica) should be in an italic form.

Response: The scientific name (Tamarindus indica) has been written in italic form (Line 46).

  1. 2. Preparation of tamarind shell extract

500 g of tamarind shell fine powder was refluxed twice with 1500 mL of 95% ethanol for 3 hours. The solvent was removed under reduced pressure. The extract was freeze-dried and stored at 4 °C.

Just wondering that why did you have to use freeze dryer after evaporation the ethanol by using the rotary evaporator as your extract contained with water only 5% (95% ethanol)!!!?

Response: Thank you for your comment. Freeze dryer was employed to remove as much water as possible.

  1. The authors should add the solvent as you used for dissolve your extract in each experiment because it is important for bioassay, sometimes the solvent can give the false positive or false negative result.

Response: Thank you for your suggestion. In the original text, we added “TSE was dissolved in ultrapure water (50 mg/mL) and stored at 4 °C. The solution was diluted to the experimental concentration with the corresponding culture medium in the following experiment.” (Line 91-93).

  1. 3. Zebrafish maintenance and exposure

I would suggest to make two subtitles as:

2.3.1 Toxicity

2.3.2 Exposure

In order to clarify the different concentration as you used in these experiments.

Response: Thank you for your suggestion. We have divided it into two sections, 2.3.1 Toxicity and 2.3.2 Exposure (Line 103; Line 109).

  1. 1. TSE protected ethanol-induced damage in HepG2 cells

The cell toxicity of TSE was tested on HepG2 cells firstly. The cell viability suggested that TSE had little toxicity to HepG2 cells with a concentration of ≤ 60 µg/mL (Figure 1A).

I think it should be…….. ³ 60 µg/mL (Figure 1A).

Response: Thank you for your suggestion. We have described it again as follows: The cell viability suggested that TSE had no significant toxicity to HepG2 cells with a concentration of less than or equal to 60 µg/mL (Figure 1A) (Line 212-213).

  1. 1. TSE protected ethanol-induced damage in HepG2 cells

We found that all doses of TSE could rescue the ethanol-treated cells from death and reverse the morphological alterations (Figure 1C and D)

How about the dose at 40 µg/mL?

Response: Thank you for your comment. In fact, TSE of 40 µg/mL also alleviated ethanol-induced cell death (p < 0.001), though the protective effect was weaker than 5-20 µg/mL.

  1. Please edit small size of the letter at Figure 5….. n =6 biologically independent….

Response: Thank you for your advice. We have added the size of biologically independent samples (Line 338).

  1. Page 13, line 409, please edit 60 µg mL-1to µg/mL

Response: We have modified it to 60 μg/mL (Line 409).

  1. References: should follow the journal format.

Response: The references have been revised to the journal format (Line 496; Line 500; Line 508-509; Line 514; Line 527; Line 542 and Line 545).

Reviewer 2 Report

The manuscript is interesting and generally well written, although there are flaws in the wording of the footnotes of figures 3 and 5.

The conclusions are very limited and do not reflect what is presented in the manuscript with the models used.

Although toxicity of the extract is ruled out, at 100 micrograms or more it can be toxic. 

In Figure 1C, the protective effect decreases at the highest dose, although the opposite was expected. The same appears in Figure 1F with the concentration of 10 micrograms/ml with respect to that of 5 micrograms.

Figure 2D does not show the standard deviation, although it is mentioned in the footer of Figure 2.

The experimental models used are interesting, although liver slices or primary hepatocyte culture (mouse or rat) would have more reliable results.

Author Response

Review 2

We are grateful for your positive comments and constructive feedback. The manuscript has been exactly revised according to your suggestions point by point. All the modifications have been highlighted in red in the revised manuscript.

Comments and response:

  1. The manuscript is interesting and generally well written, although there are flaws in the wording of the footnotes of figures 3 and 5.

Response: Thank you for your comment. We have rephrased the legends of Figures 3 and 5, and corrected some format errors (Line 282 and Line 287).

  1. The conclusions are very limited and do not reflect what is presented in the manuscript with the models used.

Response: Thank you for your comment. We have summarized the advantages of the model and its future application of TSE in conclusion (Line 465-470).

  1. Although toxicity of the extract is ruled out, at 100 micrograms or more it can be toxic.

Response: Thank you for your comment. Although the cell viability decreased to 84.71% at 100 μg/mL of TES, an excellent cytoprotective effect was observed at only 5 μg/mL of TES. Therefore, the effective dose TSE is much lower than its toxic dose.

  1. In Figure 1C, the protective effect decreases at the highest dose, although the opposite was expected. The same appears in Figure 1F with the concentration of 10 micrograms/ml with respect to that of 5 micrograms.

Response: Thank you for your comments. Although the effective dose of TSE varied sightly in each experiment, its protective effect against alcohol induced cell damage is confirmed. However, the reason for this difference is not very clear.

  1. Figure 2D does not show the standard deviation, although it is mentioned in the footer of Figure 2.

Response: Thank you for your comment. I'm sorry to make such a wrong expression. In fact, Figure 2D presents the change of DCF fluorescence intensity with time. We have modified it (Line 257-258).

  1. The experimental models used are interesting, although liver slices or primary hepatocyte culture (mouse or rat) would have more reliable results.

Response: Thank you for your comments. Chicken embryo indeed a good high throughput model for studies of efficacy and toxicity evaluation of natural products (Pharmacological Research, 2018, 133: 21-34). Of course, we will conduct a more in-depth investigation about TSE in the mouse or rat model in future.

Reviewer 3 Report

Dear Authors,

The manuscript (foods-2252089) entitled (Anti-alcohol and hepatoprotective effects of tamarind shell extract on ethanol-induced damage to HepG2 cells and animal models) has been reviewed.

The main question addressed by the research is the establishment of an ALD model on chick embryo to explore 459 the hepatoprotective activity of TSE.

The topic is original and relevant in the field. It addresses a novelty that TSE effectively attenuated ethanol-induced liver injury by relieving oxidative stress via activating the NRF2 pathway, and this  could  facilitate the utilization of TSE. 

The results are well presented and discussed.

The conclusions consistent with the evidence and arguments presented

and they address the main question.

The references appropriate, mostly new and support the discussion of the findings.

However, some comments on the MS are to be considered :

. Future prospects should be added to the conclusions 

. The choice of TSE doses administrated should be justified.

. A schematic experimental design should be added to the MS

- in figure 1 (B, C, E, F), Figure 2 (C), figure 3 (C,D), figures 4,5 and 6, the colour of histograms should be changed and different from one parameter to an other to make clear the different results obtained.

Best regards

Author Response

Review 3

We are grateful for your positive comments and constructive feedback. The manuscript has been exactly revised according to your suggestions point by point. All the modifications have been highlighted in red in the revised manuscript.

Comments and response:

  1. Future prospects should be added to the conclusions 

Response: Thank you for your suggestion. We have summarized the advantages of the model and future application of TSE in conclusion (Line 465-470).

  1. The choice of TSE doses administrated should be justified

Response: Thank you for your comment. We evaluated the toxicity of TSE in HepG2 cells and zebrafish (Figure 1A and 2A). A safe dose range was considered for the subsequent experiments. Likewise, the used dose of TSE is not harmful to chicken embryos in terms of our observation of previous experiments.

  1. A schematic experimental design should be added to the MS

Response: Thank you for your suggestion. We added a schematic illustration of our study in Figure 7 (Line 458-461).

  1. in figure 1 (B, C, E, F), Figure 2 (C), figure 3 (C,D), figures 4,5 and 6, the colour of histograms should be changed and different from one parameter to an other to make clear the different results obtained.

Response: Thanks for your excellent suggestion. We painted histograms in different colors in above mentioned figures to make our results clearer to the reader.

Reviewer 4 Report

A manuscript presents investigation on the potential of tamarind shell extract on the repression of oxidative stress induced by ethanol in larvae and chicken embryo models as well as on HepG2 cells in vitro. It describes at the appropriate manner the purpose of the investigation, techniques and methods, important data and conclusions. It gives good preliminary background and information for the possible further clinical investigations and practice as well as investigation in area of bioavailability, encapsulation techniques and identification of which component/flavonoid of extract is most important for the investigated activity.

After reviewing a manuscript I suggest minor corrections:

1. Line 19: ROS-full name for the abbreviation for the first time

2. Line 24: abbreviation for the Nrf2 is not the same throughout the text (Nrf2 or NRF2)

3. Line 207: There is no significant toxicity below concentration of 60 μg/ml, Figure 1A. It should be: Had no significant toxicity instead of ….. had little toxicity.

Author Response

Review 4

We are grateful for your positive comments and constructive feedback. The manuscript has been exactly revised according to your suggestions point by point. All the modifications have been highlighted in red in the revised manuscript.

Comments and response:

  1. Line 19: ROS-full name for the abbreviation for the first time

Response: Thanks for your comment. We have provided the full name of reactive oxygen species (ROS) (Line 19).

  1. Line 24: abbreviation for the Nrf2 is not the same throughout the text (Nrf2 or NRF2)

Response: Thanks for your comment. Due to the different forms of gene writing between different species, "Nrf2" is used in chicken, and "NRF2" is used in human (HepG2 cells) (Nrf2 appears in Line 201-203, Table 1; Line 364; Others are NRF2).

  1. Line 207: There is no significant toxicity below concentration of 60 μg/ml, Figure 1A. It should be: Had no significant toxicity instead of ….. had little toxicity

Response: Thanks for your suggestion. We have revised it (Line 212).

Round 2

Reviewer 2 Report

Although the above observations have been addressed, there are still spelling details.

to be reviewed, such as line 40 on page 1.

Figure 6A shows a code "C8DEF9" that does not correspond to the figure

or its meaning is unclear.

Author Response

Review 2

We are grateful for your positive comments and constructive feedback. The manuscript has been exactly revised according to your suggestions point by point. All the modifications have been highlighted in red in the revised manuscript.

Comments and response:

  1. Although the above observations have been addressed, there are still spelling details to be reviewed, such as line 40 on page 1.

Response: Thank you for your comment. We have checked and corrected it (Line 40).

  1. Figure 6A shows a code "C8DEF9" that does not correspond to the figure or its meaning is unclear.

Response: Thank you for your comment. We have corrected it (Line 362).
